# Seroepidemiology of Measles, Mumps and Rubella on Bonaire, St. Eustatius and Saba: The First Population-Based Serosurveillance Study in Caribbean Netherlands

**DOI:** 10.3390/vaccines7040137

**Published:** 2019-10-01

**Authors:** Regnerus A. Vos, Liesbeth Mollema, Rob van Binnendijk, Irene K. Veldhuijzen, Gaby Smits, Alcira V.A. Janga-Jansen, Sharda Baboe-Kalpoe, Koen Hulshof, Fiona R.M. van der Klis, Hester E. de Melker

**Affiliations:** 1Centre for Infectious Disease Control, National Institute for Public Health and the Environment (RIVM), Antonie van Leeuwenhoeklaan 9, 3720 MA Bilthoven, The Netherlands; 2Department of Public Health, Public Entity Bonaire, Kaya Neerlandia 41, Kralendijk Bonaire, Caribbean Netherlands, BQ-BO Kralendijk, The Netherlands; 3Department of Public Health, Public Entity St. Eustatius, Cottageroad z/n, Oranjestad, St. Eustatius, Caribbean Netherlands, 000BQ St. Eustatius, The Netherlands; 4Department of Public Health, Public Entity Saba, The Bottom, Saba, Caribbean Netherlands, 000BQ Saba, The Netherlands

**Keywords:** MMR vaccination, seroepidemiology, measles, mumps, rubella, Caribbean Netherlands, Bonaire, St. Eustatius, Saba

## Abstract

The National Immunization Program (NIP) on Bonaire, St. Eustatius and Saba (i.e., Caribbean Netherlands (CN)) includes the measles-mumps-rubella (MMR) vaccine since 1988/89. Seroepidemiological data is an important tool to evaluate the NIP, hence a cross-sectional representative population-based serosurveillance study was conducted for the first time in CN in mid-2017. Participants (*n* = 1829, aged 0–90 years) donated a blood sample and completed a health-related questionnaire. MMR-specific IgG antibodies were determined using a bead-based multiplex immunoassay and risk factors were analyzed using logistic regression models. Overall seroprevalence was high for measles (94%), but lower for mumps and rubella (both 85%). In NIP eligibles, including women of childbearing age, rubella seroprevalence (88%) exceeded the threshold for protection (85%); however, for measles (89%) this protective level (95%) was not met. MMR seropositivity was lowest in children who became CN resident at 11–17 years of age (especially for measles (72%)), mostly originating from Latin America and other non-Western countries. Interestingly, rubella seroprevalence was lowest in non-NIP eligible adults from Dutch overseas territories and Suriname (75%). Taken together, MMR immunity is generally good in CN, nonetheless some risk groups were identified. Additionally, we found evidence for a unique island epidemiology. In light of recent regional measles outbreaks, disease monitoring remains of utmost importance.

## 1. Introduction

Measles, mumps and rubella (MMR) are highly contagious viral diseases. Vaccination with trivalent MMR vaccine is safe and very effective at protecting against disease and severe complications [1]. Although incidence of MMR has declined drastically since the introduction of routine vaccination in the 1980s, elimination remains challenging, particularly for measles [2]. In fact, recent global resurgence of measles, involving large outbreaks in the World Health Organization (WHO) Region of the Americas, is of great concern as vaccination coverage is frequently insufficient to achieve herd protection in most countries [3,4].

Caribbean Netherlands (CN), situated in the Caribbean Sea, consists of three Dutch special municipalities: Bonaire (one of the Dutch Leeward Antilles together with Aruba and Curaçao), St. Eustatius and Saba (both 800 km to the northeast). MMR vaccinations have been administered routinely in CN for decades. Appendix A gives an overview of introduction of MMR vaccinations and adaptations since 1975. Currently, the National Immunization Program (NIP) recommends two doses against MMR: On Bonaire, the first dose (MMR-1) is administered at 14 months and MMR-2 at 18 months (MMR-2 before 2019 at nine years of age); on St. Eustatius and Saba, MMR-1 is given at 12 months and MMR-2 at four years of age (MMR-2 before 2007 and 2016, respectively, at nine years of age) [5]. Vaccination coverage in CN has been registered routinely since a few years: In 2017, MMR-1-coverage was 92% (range 90–100%) and MMR-2 70% (range 67–100%) [6].

Since the implementation of syndromic surveillance in 2007, no imported or endemic MMR cases have been detected in CN. Additionally, registers on St. Eustatius and Saba indicated that no confirmed cases of measles or (congenital) rubella (syndrome) have occurred since the introduction of the MMR vaccination (1988). However, it should be noted that only few suspected cases undergo laboratory confirmation due to a lack of facilities. On Curaçao, outbreaks of rubella have been reported in 1977 and 1985/1986; however, its scale and dissemination to Bonaire remains unspecified. Moreover, one imported case of measles was confirmed in May 2019 on Curaçao, and Aruba reports a few confirmed cases of mumps every year [7].

Seroepidemiological data play a crucial role in profiling population immunity, and is an important tool to evaluate the NIP and, if needed, adapt its policy [8]. The recent large measles outbreak across the Americas emphasizes the urgent need for information on protection against vaccine-preventable diseases, which is lacking for CN [3]. The aim of this cross-sectional population-based seroepidemiological study was to investigate the humoral immunity against MMR in the general population of CN, which enables identification of possible gaps in immunity (seronegativity) and risk factors associated with these gaps.

## 2. Materials and Methods

### 2.1. Study Design and Study Population

From May–June 2017, a biobank was established in CN: Health Study Caribbean Netherlands. Details on study design and data collection have been described elsewhere [9]. Briefly, on each island an age-stratified sample, with age strata 0–11, 12–17, 18–34, 35–59 and 60–89 years, was randomly drawn from the population registry of the Dutch overseas territories (PIVA-V, January 1, 2017). In total, 7768 eligible individuals were invited (Bonaire *n* = 4667; St. Eustatius *n* = 2062 and Saba *n* = 1039; see Appendix A for a flowchart of the study). Prior to participation, signed informed consent was obtained (from: <12 years of age: Parent/legal guardian; 12–17 years of age: Participant and parent/legal guardian and ≥18 years of age: Participant). The study was conducted in accordance with the Declaration of Helsinki, and the protocol was approved by the Medical Ethics Committee Noord-Holland (METC-number: M015-022). At the clinic, participants were requested to donate a blood sample—via a finger or heel prick using the dried blood spot method (DBS) on air-dried filter paper (Whatman^®^ 903 protein saver cards)—to complete a questionnaire, and to bring their vaccination certificate. If the latter was not available, vaccination status was retrieved from the local public health department if obtainable.

### 2.2. Laboratory Analyses

After the fieldwork, blood samples were air-shipped to the laboratory of the National Institute for Public Health and the Environment the Netherlands (RIVM) and stored instantly at −80 °C until analyses. MMR-specific IgG antibodies were determined with a fluorescent bead-based multiplex immunoassay using Luminex technology, as described previously [10]. In short, following standard protocol, a 3.2 mm (1/8 inch) punch was taken from the DBS and incubated in 300 µL phosphate-buffered saline containing 0.1% Tween-20 and 3% bovine serum albumin (i.e., assay buffer) at 4 °C overnight on a shaker to release serum (resulting in a 1:200 dilution) [11,12]. Sera were further diluted to 1:4000 in assay buffer. Controls, blanks and the international standard for rubella (RUBI-1-94), which was calibrated against the international standard for measles and an in-house standard for mumps, were included on each plate. Antibody concentrations were obtained by interpolation of the mean fluorescent intensity in the reference serum curve using a logistic-5PL regression type and expressed in international units per mL (IU/mL) for measles and rubella and RIVM units per mL (RU/mL) for mumps—as no international standard is available. An antibody concentration of ≥0.120 IU/mL for measles [13] and ≥10.0 IU/mL for rubella [14] was considered protective and used as cut-off for seropositivity. For mumps, no correlate of protection is available: An antibody concentration of ≥45.0 RU/mL was used as arbitrary criterion for seroprevalence, upon agreement by the European Sero-Epidemiology Network [15].

### 2.3. Data Analyses

#### 2.3.1. Seroprevalence and GMC

Data were analyzed in SAS v.9.4 (SAS Institute Inc., USA) and R v.3.6. Analyses took account of the survey design: To match the population distribution on each island as of January 1, 2017, overall seroprevalence and geometric mean concentrations (GMC) for IgG antibodies were estimated by linear weighting, taking into account sex, age group and country of birth (and neighborhood on Bonaire). Differences in seroprevalence of MMR-specific antibodies between islands and gender were determined by estimating the parameters of the beta distribution for these seroprevalence rates using the methods of moments [16]. Risk ratios, their corresponding 95% confidence intervals (CI) and *p* values were estimated by Monte Carlo simulations of these seroprevalence estimates. Dissimilarities in GMC between islands and gender were identified by calculating the difference in natural logarithmic (ln) concentrations and tested using a *t*-test. Age-specific seroprevalence, GMC and 95% CI were calculated for CN and per island. *p* values of <0.05 were considered statistically significant.

#### 2.3.2. Waning Immunity after MMR Vaccination

Linear regression analyses of MMR ln-antibody concentrations were conducted to study the persistence of antibodies after one and two MMR vaccination(s) received in the Dutch overseas territories. Analyses were restricted to participants who had received MMR-1 between 13–16 months of age (as vaccine response and waning of antibodies is shown to be different in children up to 12 months of age [17]) and MMR-2 between 8–10 years of age, both given at least 2 months before inclusion in the study. Additionally, maximum age at study inclusion was 9 and 30 years for one and two dose(s), respectively. For mumps, those with self-reported mumps symptoms in the preceding year were excluded.

#### 2.3.3. Risk Factors for Seronegativity

Risk factors for MMR seronegativity were identified using separate logistic regression models. A complete case analysis was conducted for both mumps and rubella (*n* = 1816; Appendix A). Allowing the measles model to converge, those born before introduction of routine vaccination were exclude (*n_total_* = 1075), i.e., a period characterized by widespread measles circulation causing nearly all participants to be seropositive (on Bonaire from 42 years of age, and on St. Eustatius and Saba from 36 years). Information on the history of MMR vaccinations on Bonaire before 1988 was derived from neighboring Dutch Leeward Antilles island Curaçao as the same NIP was applied. Studied risk factors included sociodemographics, vaccination history and other health-related factors. Aside from age and sex, variables with *p* < 0.10 in univariate analyses were included in the multivariate analyses. Backward selection was used to identify independent determinants in which *p* < 0.05 was considered statistically significant associated. Crude and adjusted odds ratios (OR) and 95% CIs were estimated as well as unadjusted seroprevalence and 95% CIs for all studied factors.

## 3. Results

### 3.1. Study Characteristics

In the present study, 1900 participants (response rate 24.5%) were included, of which 1829 donated a blood sample (Appendix A; 824 (45%) men and 1005 (55%) women; aged 3 months to 90 years), with equal distribution over the islands according to population size (Bonaire: 1129 (62%); St. Eustatius: 477 (26%); Saba: 223 (12%); Table 1). Most participants originated from the Dutch overseas territories (comprising CN, Aruba, Curaçao and St. Maarten) and Suriname (DOT-Sur; *n* = 1312, 72%), followed by Latin America and other non-Western countries (LA-nonW; *n* = 281, 16%), and indigenous Dutch and other Western countries (iD-Wes; *n* = 223, 12%). Almost half of the participants reported to be low educated (*n* = 883), compared to 26% middle and 18% high (8% unknown). On Saba, relatively more iD-Wes, LA-nonW and those with a high educational level participated—consistent with its population composition [18,19]—as compared to Bonaire and St. Eustatius. Among NIP eligible participants, i.e., those born in the MMR vaccination era, registered vaccination coverage with at least one dose against MMR ranged between 69–76% among the islands, and 8–9% were unvaccinated (and the remainder self-reported to have (partly) participated in the NIP).

### 3.2. Age-Specific Seroprevalence and GMC

Table 2 shows the overall weighted IgG seroprevalence and GMC of MMR in the total CN population, stratified by island and sex, and among NIP eligibles and non-NIP eligible adults. In total, 72.0% (*n* = 1337) were seropositive for all three pathogens, and 2.5% seronegative (*n* = 47; of which *n* = 26 had not reached the NIP eligible age, including all infants between 3–5 months of age (*n* = 6) for whom protective maternal antibody concentrations could be expected). There was no difference in overall seroprevalence for MMR between islands (all *p* > 0.05). Weighted MMR seroprevalence and GMC, stratified by age groups, for CN are depicted in Figure 1A–C, and per island in Appendix A. The possible effect of storage and transportation on antibody concentrations of DBS samples in this study was investigated: No significant difference (all *p* > 0.05, one-way ANOVA) was found between MMR antibody concentrations of samples obtained at the start of the study (having the longest storage period (4 weeks)) compared to samples stored shorter (1, 2 and 3 weeks)—while displaying a similar age distribution.

#### 3.2.1. Measles

Overall weighted measles IgG seroprevalence was 93.8% (95% CI 92.3–95.2), with an overall GMC of 0.93 IU/mL (95% CI 0.86–1.01; Table 2). Seroprevalence in CN did not differ significantly between men and women (93.1% vs. 94.5%, respectively, *p* = 0.31); however overall GMC was lower for men (0.87 vs. women: 1.00 IU/mL, *p* = 0.03). On St. Eustatius a sex difference was observed in both the total population (seroprevalence men: 91.0% vs. women: 97.0%, *p* = 0.02; and GMC men: 0.75 vs. women: 1.27 IU/mL, *p* < 0.0001) and among NIP eligibles (seroprevalence men: 81.4% vs. women: 93.3%, *p* = 0.01; data not shown).

Seroprevalence in CN increased rapidly from 64.7% at one year of age to 94.2% at two years, with a corresponding upsurge of GMC (from 0.27 to 1.39 IU/mL, respectively) reflecting the vaccine response to MMR-1 at 12/14 months of age (Figure 1A). Seroprevalence fluctuated between 85–100% for children up to 18 years of age, with the lowest seropositivity among residents from LA-nonW (e.g., 65.5% in age group 12–17 years; Figure 2A). GMC declined to 0.37 IU/mL at seven years of age, from where it remained range bound between 0.30–0.56 IU/mL until 35 years (Figure 1A). In adults aged 18–35 years, seroprevalence ranged between 80–95%. Seroprevalence and GMC were lower for NIP-eligible participants (89.2% and 0.46 IU/mL, respectively) as compared to non-NIP eligible adults (98.6% and 1.96 IU/mL, respectively; Table 2). From 35 years of age, GMC rapidly inclined to 2.5 IU/mL at 60 years, and seroprevalence remained 100% from 55 years onward (Figure 1A).

#### 3.2.2. Mumps

Overall weighted mumps IgG seroprevalence was 85.0% (95% CI 83.0–87.0)—with an overall GMC of 125 RU/mL (95% CI 133–188)—and was similar between NIP eligibles and non-NIP eligible adults (Table 2). GMC was lower on St. Eustatius (104 RU/mL) as compared to Bonaire and Saba (both *p* < 0.005). Among sexes, overall seroprevalence and GMC were similar (*p* > 0.05; Table 2); however, on St. Eustatius seroprevalence (men: 76.2% vs. women: 86.1%, *p* = 0.03) and GMC (men: 85.8 vs. women: 127 RU/mL, *p* < 0.0001) were higher in women (data not shown). Symptoms of mumps in the preceding year (*n_total_* = 27) were most frequently self-reported by 13–34 year-olds (*n* = 16), and their GMC was higher than those 13–34 years without symptoms (202 vs. 116 RU/mL, respectively, *p* = 0.02).

Seroprevalence in CN was 74.1% at two years of age (after MMR-1) and rose to 93.2% at 5 years, with a corresponding increase in GMC from 37 to 197 RU/mL, respectively, reflecting the response to MMR-2 at 4 years on St. Eustatius and Saba (Figure 1B and Appendix A). At 10 years of age (after MMR-2 on Bonaire), seroprevalence in CN was 89.2% and GMC 154 RU/mL. Thereafter, seroprevalence and GMC fluctuated between 80–94% and 106–169 RU/ml, respectively, with age group 18–20 years displaying the highest prevalence (94.4%). All islands showed a similar trend in seroprevalence with age, except for age group 18–34 years in which seroprevalence was considerably higher on Bonaire (90.2%) than the other islands (<70%; Appendix A). Overall seroprevalence in non-NIP eligible adults was lowest in residents from DOT-Sur (82.2%, and, e.g., in age group 60–90 years: 85.3% vs. 92.8% in iD-Wes and 88.6% in LA-nonW; Figure 2B)).

#### 3.2.3. Rubella

Overall weighted rubella IgG seroprevalence was 84.5% (95% CI 82.4–86.6)–with a GMC of 31.2 IU/mL (95% CI 34.2–28.5)—and was higher among NIP eligibles (87.5%) as compared to non-NIP eligible adults (80.6%, *p* = 0.002; Table 2). GMC on St. Eustatius was lower than on the other islands (24.8; Bonaire: 32.0 and Saba: 36.6 IU/mL (St. Eustatius vs. Bonaire: *p* = 0.02 and St. Eustatius vs. Saba: *p* = 0.004)). Between sexes, no difference in overall seroprevalence was observed in CN (men: 86.0% vs. women 83.0%, *p* = 0.17) and on each island (Table 2); yet, on St. Eustatius, GMC was higher in women (29.9 vs. men: 20.6 IU/mL, *p* = 0.01; data not shown). Notably, on Bonaire, seroprevalence in non-NIP eligible men (86.6%) was higher than in women (73.9%, *p* = 0.005; data not shown), also reflected by a higher overall GMC in men (36.0 vs. women: 28.2 IU/mL, *p* = 0.007; Table 2).

Seroprevalence of rubella showed a similar age pattern as measles among NIP eligibles (Figure 1C) and was consistent across the islands (Appendix A). After MMR-1, seroprevalence in CN was 94.2% for two-year-olds and fluctuated between 87–100% until 18 years of age. Participants from LA-nonW aged 12–17 years were least seropositive (79.5%; Figure 2C). GMC reached its highest peak at four years of age at 76.0 IU/mL and declined to 33.3 IU/mL at seven years (Figure 1C). From there it steadily declined towards 40 years of age (ranging between 25–39 IU/mL), reaching its lowest concentration at age group 40–44 years (19.8 IU/mL), but still above the cut-off for protection. Seroprevalence varied between 83–91% in adults aged 18–40 years. Unlike measles, seroprevalence remained indifferent after 40 years of age, varying between 76–87% and GMC between 25–47 IU/mL. Interestingly, seroprevalence and GMC in non-NIP eligible adults from DOT-Sur and LA-nonW were substantially lower than in iD-Wes (e.g., overall seroprevalence was 75.4%, 81.8% and 97.0%, respectively), and this also applied to the different age groups among them (Figure 2C)).

### 3.3. Waning Immunity after MMR Vaccination

For measles and rubella, waning of IgG antibody concentration after MMR-1 (*n* = 128) was significantly faster than after MMR-2 (*n* = 126; all *p* < 0.05), namely for measles with a slope of –0.25 and –0.04 ln-IU/mL per year, respectively; and for rubella with a slope of –0.19 and –0.04 ln-IU/mL per year, respectively (Appendix A). For mumps, no decline in antibody concentration was observed after MMR-1 (*n* = 125; slope: 0.03 ln-RU/mL per year, *p* = 0.55,) and MMR-2 (*n* = 124; slope: –0.02, *p* = 0.31; Appendix A).

### 3.4. Risk Factors for Seronegativity

Risk factors for measles were solely studied among NIP eligible participants as non-NIP eligible adults were nearly all seropositive. In multivariate analysis, men (vs. women), infants aged 0–1 years (vs. 2–10), those who have been resident of CN since age group 11–17 years (vs. 0–1) and participants who self-reported to have (partly) followed the NIP and who were unvaccinated (vs. two or more doses) had significantly higher odds of being seronegative (Table 3). For mumps, participants aged 0–1 and 2–10 years (vs. 11–17), those who have been resident of CN since age group 11–17 years (vs. 0–1), individuals who were vaccinated once, those self-reported to have (partly) followed the NIP, those who were unvaccinated and who were not eligible for the NIP (vs. two or more doses) were found to be significant risk factors for seronegativity in multivariate analysis (Table 4). The multivariate model for rubella revealed that all age groups except 11–17 years (vs. 2–10), those who have been resident of CN since age 0–17 years (vs. 40–59), people who self-reported to have (partly) followed the NIP, those unvaccinated and who were not eligible for the NIP (vs. two or more doses) were significantly associated with seronegativity (Table 5).

## 4. Discussion

This cross-sectional population-based serosurveillance study estimated the level of humoral immunity against MMR and risk factors associated with seronegativity in CN. Overall seroprevalence was high for measles (94%), but lower for mumps and rubella (both 85%). In NIP eligibles, including women of childbearing age, rubella seroprevalence (88%) exceeded the threshold for protection (85%); however, for measles (89%) this level (95%) was not met [20,21]. MMR seropositivity was lowest in children who became CN resident at 11–17 years of age (especially for measles (72%)), mostly originating from Latin America and other non-Western countries. MMR vaccinations elicited good antibody responses and receiving two doses of MMR (vs. one) indicated prolonged humoral immunity. Interestingly, rubella seroprevalence was lowest in non-NIP eligible adults from DOT-Sur (75%), illustrative of a specific island epidemiology in the pre-vaccination era.

Overall seroprevalence for measles in CN (94%) was consistent with our previously reported estimate for Bonaire [22]. As reviewed by Dimech and colleagues [23], other large population studies reported measles seroprevalence rates between 54–96% (e.g., Italy: 74%, USA: 93%), of which the Netherlands was among the highest (96%). Seroprevalence for mumps in CN (85%) was rather similar to the USA (88%) [24], but somewhat lower than the Netherlands (91%) [25]. Rubella seroprevalence in CN (85%) was mostly lower than studies performed elsewhere, e.g., Colombia (89%), the Netherlands (95%) and Thailand (98%) [23]. Main drivers coinciding with differences in seroprevalence and antibody responses between and within populations can be attributed to vaccination status (and vaccine effectivity in general) as well as (past) natural exposure to these pathogens.

Children who reside in CN since age 11 years, i.e., after the regular NIP, had a high likelihood of being MMR seronegative. Indeed, lowest seroprevalence was observed in LA-nonW residents aged 12–17 years (e.g., for measles 66%) as they were less vaccinated. Hence, they probably missed vaccination opportunities in their country of birth due to lack of goods or migration—as their beliefs on vaccination (e.g., anti-vaccination) were indifferent from their peers (data not shown)—and did not catch up on missed vaccinations upon arrival to CN (which is regular policy). Based on these findings and in light of recent dissemination of measles across the region and influx of refugees [4], vaccination policy with respect to eligible immigrants aged <18 years was tightened where possible.

Moreover, male sex was an independent determinant for measles seronegativity among NIP eligibles (note: Additional risk factor analyses for rubella and mumps among NIP eligibles revealed a similar—although non-significant—association with sex (data not shown)). This sex difference was most prominent on St. Eustatius, while according to our registry vaccination coverage was even somewhat higher in men. To note, although vaccination status was available from a large proportion of participants, not all records could be retrieved. In that case, we used a self-reported variable on overall NIP attendance as a surrogate, which could not differentiate between (number of) vaccines unfortunately, and recall bias might have played a role too. Hence, we could not exclude that vaccination coverage against MMR on St. Eustatius was slightly higher in women as compared to men. Conversely, given the higher GMC in women too, women might serologically respond better to the vaccine components, as postulated by others [26,27]. Nonetheless, as cellular immunity is assumed to be an essential part of protection [28], higher risk of susceptibility among vaccinated men remains questionable. Future research in outbreak settings might provide clarification on this potential sex difference in vaccinated cases.

We detected significant dissimilarities between NIP eligibles and non-NIP eligible adults. Generally, the latter have (frequently) been naturally exposed to MMR during their life, elucidating high GMCs, indicative of lifelong protection [25,29,30]. This concept is best underlined by measles, a highly infectious agent that is capable of disseminating throughout susceptible populations [31]. Hence, nearly all non-NIP eligible adults in our study were seropositive for measles displaying high antibody concentrations, i.e., were infected, possible boosted regularly and thus protected. However, this was different for the less infectious pathogens mumps and, in particular, rubella. Interestingly, adult participants who were born on the islands or resided there since childhood were more likely to be seronegative. This was confirmed by a lower seroprevalence and GMC in adults from DOT-Sur and LA-nonW descent when compared to iD-Wes who were born in rubella endemic countries (mostly the Netherlands) prior to introduction of MMR vaccination. Principally for rubella, differences in seroprevalence between countries in the pre-vaccination period have been described [32]. Hence, as CN was even more remote and isolated during the pre-globalization/vaccine era, we hypothesize that introduction and transmission of rubella occurred less often due to its lesser infectious character, causing less circulating and exposure, affecting less people. Whilst a proportion of these inhabitants might still be susceptible currently and future cases cannot be ruled-out completely, disease in elderly is expected to be mostly mild, and yet sufficient herd immunity should prevent transmission. Fortunately, seroprotection for rubella was above the threshold for protection in NIP eligibles, including women of childbearing age who are at risk of developing congenital rubella syndrome—resulting in serious birth defects or miscarriage—via infection with rubella during pregnancy [20].

Consistent with literature, waning immunity of measles and rubella specific IgG antibodies after vaccination was present, but much slower after a second dose, staying well above the cut-off for seropositivity [29,30,33]. This indicates long lasting immunological humoral memory—when extended with similar rate of waning. Although this underlines the purpose of booster vaccination—besides preventing primary vaccine failure—we could not draw firm conclusions on the persistence of these antibodies as these data were cross-sectional. Similarly, the non-prospective character of our data was most likely the reason why mumps antibody levels were indifferent eight years after the first dose. Furthermore, seroprevalence rates for mumps should be interpreted with caution as a defined correlate of protection—albeit recent research endeavored [34]—is still lacking. While two doses of MMR (vs. one) indicated a lower risk of mumps seronegativity in our study, outbreaks among twice vaccinated students—with intensive and homogenous contact—have been reported [35]. In fact, in contrast to St. Eustatius and Saba, we observed a high seroprevalence and GMC in young adults on Bonaire and self-reporting of mumps symptoms was highest among this group too. This, together with confirmed cases from nearby Dutch Leeward Antilles island Aruba and recurrent traffic between these islands, could suggest possible exposure to mumps on Bonaire, whereas this likelihood might be lower on St. Eustatius and Saba, which are more isolated.

All infants too young to be vaccinated with MMR were seronegative in our study; even among the infants 3–5 months of age for whom protective maternal antibody concentrations could be expected. This phenomenon is well-known among babies from mothers who have not been naturally infected as antibody concentrations from vaccination are significantly lower and thus reach cutoffs for seropositivity earlier [36]. Timely vaccination and close monitoring remains of great importance, especially in light of recent regional circulation of the measles virus and migration of large populations at risk (e.g., from Venezuela) [3,4]. While considering an optimal age for vaccination, health authorities should take into account several factors, including immunological response, vaccine coverage, herd immunity thresholds and risk of infection [37]. Although no cases of measles have been detected via the surveillance systems in CN recently, the public health department on Bonaire decided to lower the age of MMR-2 from 9 years to 18 months of age (as of January 1, 2019), in order to timely and adequately protect young infants and reduce the risk of viral introduction and transmission.

This study has some additional limitations. Due to the overall response rate of 25%, the possibility of non-response bias cannot be excluded (as described earlier [9]). However, we partly corrected for this by weighting our sample on important sociodemographic characteristics. Further, to overcome logistical hurdles, we used DBS to collect our blood samples. Whilst this is a widely used and validated method for measuring antibodies, we could not exclude that storage and transportation might have had some effect on the antibody levels. We have investigated this and found little overall effect that has not affected our results.

## 5. Conclusions

In conclusion, this is the first large-scale serosurveillance study in CN providing evidence on humoral immunity against MMR. The CN population is overall well protected against MMR, albeit some groups were identified that could be at risk of infection. Our data also indicate that infectious disease epidemiology on these islands might have been different in the pre-vaccination era as compared to past MMR endemic countries, such as the Netherlands. Particularly in light of recent outbreaks in the region, it is important to have sensitive disease surveillance in place and to sustain high vaccination coverage in order to meet herd immunity thresholds and, ultimately, reach the WHO measles and rubella elimination goals [20]. Lastly, it is highly recommended to conduct serosurveillance studies in CN on a regular basis in the future in order to monitor the protection against vaccine-preventable diseases and timely detect (additional) gaps in population immunity [8].

## Figures and Tables

**Figure 1 vaccines-07-00137-f001:**
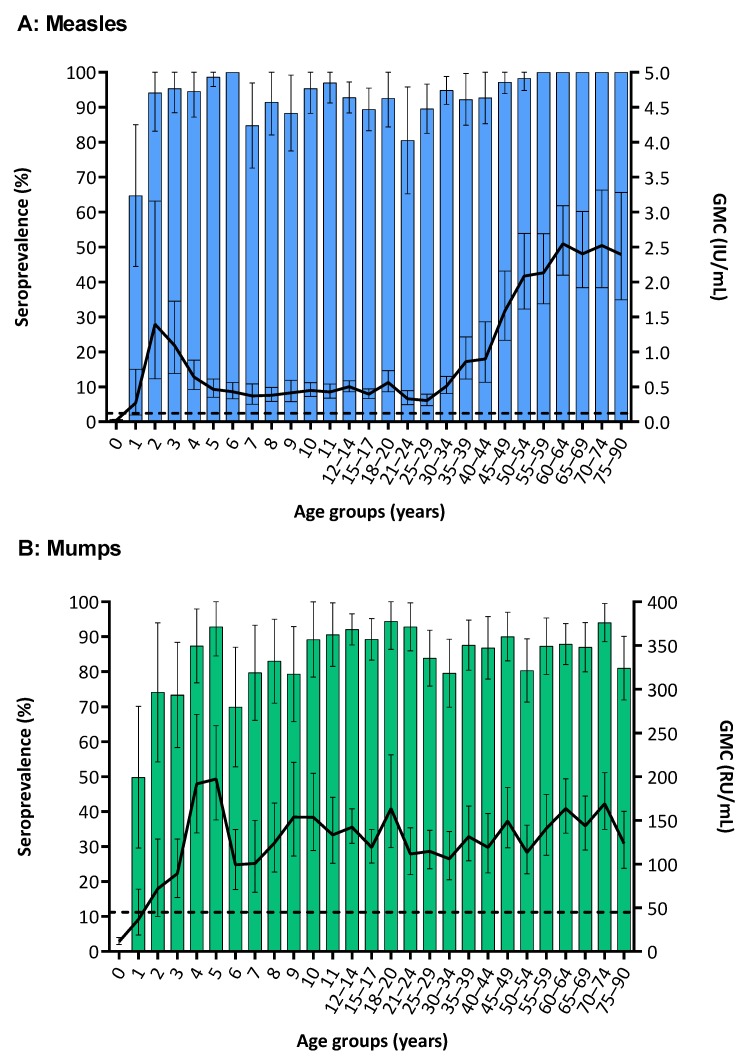
Age-specific seroprevalence (%) and geometric mean concentration (GMC) (with 95% confidence intervals) of measles (**A**), mumps (**B**) and rubella (**C**) IgG antibodies in the general population of Caribbean Netherlands, 2017. Note: Antibody concentrations ≥0.120 international units (IU)/mL for measles, ≥45.0 RIVM units (RU)/mL for mumps and ≥10.0 IU/mL for rubella were considered seropositive (dashed lines).

**Figure 2 vaccines-07-00137-f002:**
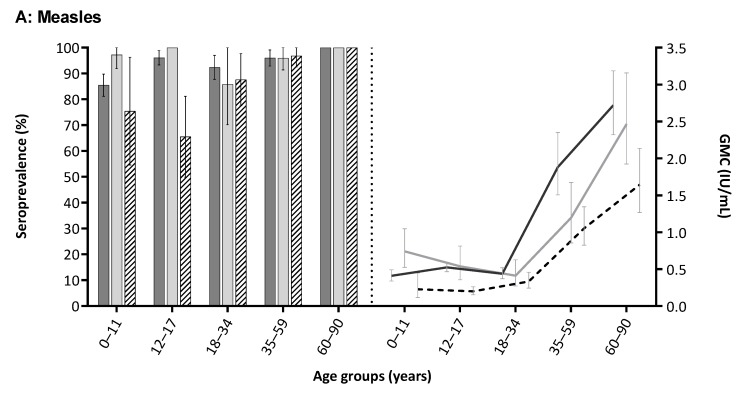
Age-specific seroprevalence (%) and geometric mean concentration (GMC) (with 95% confidence intervals) of measles (**A**), mumps (**B**) and rubella (**C**) IgG antibodies in the general population of Caribbean Netherlands, 2017, by ethnicity.

**Table 1 vaccines-07-00137-t001:** Sociodemographic characteristics and vaccination history of participants with a blood sample in the Health Study Caribbean Netherlands, by island.

Sociodemographic Characteristics and Vaccination History		Bonairen *n* (%) *n* = 1129 (61.7)	St. Eustatiusn *n* (%) *n* = 477 (26.1)	Saban *n* (%) *n* = 223 (12.2)	Totaln *n* (%) *n* = 1829
**Sex**					
	Men	506 (44.8)	221 (46.3)	97 (43.5)	824 (45.1)
	Women	623 (55.2)	256 (53.7)	126 (56.5)	1005 (54.9)
**Age, mean (sd)**		34.6 (25.0)	30.8 (23.7)	37.5 (25.3)	34.0 (24.8)
**Age groups, years**					
	0–11	271 (24.0)	128 (26.8)	50 (22.4)	449 (24.6)
	12–17	181 (16.0)	86 (18.0)	24 (10.8)	291 (15.9)
	18–34	160 (14.2)	83 (17.4)	32 (14.3)	275 (15.0)
	35–59	242 (21.4)	99 (20.8)	60 (26.9)	401 (21.9)
	60–90	275 (23.4)	81 (17.0)	57 (25.6)	413 (22.6)
**Ethnicity ^a^**					
	Dutch overseas territories and Suriname	803 (71.2)	383 (82.0)	126 (57.0)	1312 (72.2)
	Indigenous Dutch and other Western countries	143 (12.7)	30 (6.4)	50 (22.6)	223 (12.3)
	Latin America and other non-Western countries	182 (16.1)	54 (11.6)	45 (20.4)	281 (15.5)
**(Maternal) educational level ^b^**					
	High	172 (15.2)	68 (14.3)	87 (39.0)	327 (17.9)
	Middle	298 (26.4)	125 (26.2)	45 (20.2)	468 (25.6)
	Low	571 (50.6)	232 (48.6)	80 (35.9)	883 (48.3)
	Unknown	88 (7.8)	52 (10.9)	11 (4.9)	151 (8.2)
**Monthly gross income**					
	High (≥$3001)	197 (17.4)	91 (19.1)	60 (26.9)	348 (19.0)
	Middle ($1501–3000)	328 (29.1)	88 (18.5)	60 (26.9)	476 (26.0)
	Low (<$1500)	329 (29.1)	133 (27.8)	56 (25.1)	518 (28.3)
	Does not want to answer	106 (9.4)	73 (15.3)	23 (10.3)	202 (11.1)
	Unknown	169 (15.0)	92 (19.3)	24 (10.8)	285 (15.6)
**Vaccination history among National Immunization Program (NIP) eligible participants ^c^**					
**Measles, total**		672 (59.5)	302 (63.3)	107 (48.0)	1081 (59.1)
	2 or more doses	215 (32.0)	106 (35.1)	29 (27.1)	350 (32.4)
	1 dose	248 (36.9)	118 (39.1)	51 (47.7)	417 (38.6)
	(Partly) participated in the NIP (self-reported)	148 (22.0)	47 (15.5)	20 (18.7)	215 (19.9)
	Not vaccinated	61 (9.1)	31 (10.3)	7 (6.5)	99 (9.1)
**Mumps, total**		624 (55.3)	263 (55.1)	106 (47.5)	993 (54.3)
	2 or more doses	213 (34.1)	99 (37.6)	29 (27.4)	341 (34.3)
	1 dose	245 (39.3)	113 (43.0)	51 (48.1)	409 (41.2)
	(Partly) participated in the NIP (self-reported)	115 (18.4)	30 (11.4)	19 (17.9)	164 (16.5)
	Not vaccinated	51 (8.2)	21 (8.0)	7 (6.6)	79 (8.0)
**Rubella, total**		736 (65.2)	263 (55.1)	106 (47.5)	1105 (60.4)
	2 or more doses	216 (29.3)	100 (38.0)	29 (27.4)	345 (31.2)
	1 dose	249 (33.8)	112 (42.6)	51 (48.1)	412 (37.3)
	(Partly) participated in the NIP (self-reported)	197 (26.8)	30 (11.4)	19 (17.9)	246 (22.3)
	Not vaccinated	74 (10.0)	21 (8.0)	7 (6.6)	102 (9.2)

^a^ Dutch overseas territories include the islands: Bonaire, Saba and St. Eustatius (i.e., Caribbean Netherlands), and Aruba, Curaçao and St. Maarten. Within the ethnic group of indigenous Dutch and other Western countries, *n* = 147 (66%) were indigenous Dutch. Within the ethnic group of Latin America and other non-Western countries, *n* = 261 (93%) were born in Latin America. ^b^ Maternal educational level was used for participants 0–11 y, active education was used for participants 12–25 y and highest accomplished educational level was used for participants >25 y. Low = no education, primary school, pre-vocational education (VMBO), lower vocational education (LBO/MBO-1) and lower general secondary education (MAVO/VMBO). Middle = intermediate/secondary vocational education (MBO-2-4), higher/senior vocational education (HAVO) and pre-university education (VWO/Gymnasium); high = higher professional education (HBO), university BSc., university MSc. and doctorate. ^c^ On Bonaire, NIP eligible participants for measles include those until 41 y, for mumps 36 y and for rubella 52 y for women and 44 y for men (in accordance with data from Curaçao). On St. Eustatius NIP eligible participants for measles include those until 35 y, and for mumps and rubella 29 y. On Saba NIP eligible participants for measles include those until 35 y, and for mumps and rubella 34 y. The self-reported variable on NIP participation was used if a vaccination certificate was unavailable. A participant was categorized as ‘not vaccinated’ if both a vaccination certificate was unavailable as well as if they self-reported about no participation in the NIP or did not know whether they participated. Missing: Ethnicity *n* = 13.

**Table 2 vaccines-07-00137-t002:** Weighted seroprevalence (%) and geometric mean concentration (GMC) (with 95% confidence intervals) of measles, mumps and rubella IgG antibodies in the national population of Caribbean Netherlands.

		Measles				Mumps				Rubella			
		Seroprevalence ≥0.120 IU/mL		GMC		Seroprevalence ≥45 RU/mL		GMC		Seroprevalence ≥10.0 IU/mL		GMC	
		%	(95% CI)	IU/mL	(95% CI)	%	(95% CI)	RU/mL	(95% CI)	%	(95% CI)	IU/mL	(95% CI)
**Total Caribbean Netherlands population**													
Overall		93.8	(92.3–95.2)	0.93	(0.86–1.01)	85.0	(83.0–87.0)	125	(133–188)	84.5	(82.4–86.6)	31.2	(28.5–34.2)
**Island**													
	Bonaire	93.7	(91.9–95.4)	0.92	(0.83–1.02)	86.0	(83.7–88.3)	129	(120–138)	85.1	(82.6–87.6)	32.0	(28.7–35.6)
	St. Eustatius	93.9	(91.2–96.5)	0.97	(0.83–1.14)	81.0	(76.2–85.8)	104	(92–118)	82.3	(77.8–86.8)	24.8	(20.3–30.2)
	Saba	94.9	(91.4–98.4)	1.01	(0.81–1.24)	81.4	(75.4–87.4)	135	(113–161)	82.7	(76.8–88.6)	36.6	(27.5–48.7)
**Sex**													
	Men	93.1	(90.8–95.3)	0.87	(0.76–0.99)	84.9	(81.8–87.9)	120	(110–131)	86.0	(82.8–89.1)	33.6	(29.2–38.6)
	Women	94.5	(92.8–96.2)	1.00	(0.90–1.12)	85.1	(82.5–87.7)	131	(121–142)	83.0	(80.2–85.8)	28.8	(25.6–32.5)
**Among NIP eligibles ^a^**													
Overall		89.2	(86.7–91.8)	0.46	(0.42–0.51)	83.9	(81.0–86.8)	116	(107–125)	87.5	(84.8–90.1)	30.6	(27.5–34.0)
**Among non-NIP eligible adults**													
Overall		98.6	(97.6–99.7)	1.96	(1.77–2.18)	85.6	(83.0–88.7)	133	(123–145)	80.6	(77.1–84.1)	32.1	(27.3–37.7)

^a^ On Bonaire, NIP eligible participants for measles include those until 41 y, for mumps 36 y and for rubella 52 y for women and 44 y for men (in accordance with data from Curaçao). On St. Eustatius, NIP eligible participants for measles include those until 35 y, and for mumps and rubella 29 y. On Saba, NIP eligible participants for measles include those until 35 y, and for mumps and rubella 34 y. Abbreviations: CI, confidence interval; IU/mL, international units per mL; RU/mL, RIVM units per mL.

**Table 3 vaccines-07-00137-t003:** Risk factor analysis for measles IgG seronegativity among the Health Study Caribbean Netherlands population without non-National Immunization Program (NIP) eligible adults ^a^.

Potential Risk Factor for Measles Seronegativity		*n* (%) *n* = 1075	% Measles Seropositive (95% CI)	UnivariateCrude OR ^b^ (95% CI)	*p* Value ^c^	MultivariateaOR ^b^ (95% CI)	*p* Value ^c^
**Island**					**0.04**		
	Bonaire	671 (62.4)	89.4 (87.1–97.7)	Ref.			
	St. Eustatius	297 (27.6)	92.3 (89.2–95.3)	0.62 (0.37–1.06)			
	Saba	107 (10.0)	94.4 (90.0–98.8)	**0.38 (0.15–0.95)**			
**Sex**					**0.01**		**0.003**
	Men	492 (45.8)	88.6 (85.8–91.4)	**1.78 (1.14–2.78)**		**2.06 (1.29–3.30)**	
	Women	583 (54.2)	92.5 (90.3–94.6)	Ref.		Ref.	
**Age group, years**					**<0.0001**		**<0.0001**
	0–1	49 (4.6)	49.0 (35.0–63.0)	**17.94 (8.71–36.99)**		**8.78 (3.80–20.27)**	
	2–10	356 (33.1)	94.1 (91.6–96.6)	Ref.		Ref.	
	11–17	335 (31.1)	93.4 (90.8–96.1)	1.13 (0.61–2.11)		0.54 (0.26–1.15)	
	18–29	172 (16.0)	88.4 (83.6–93.2)	**2.29 (1.20–4.37)**		1.08 (0.50–2.32)	
	30–41	163 (15.1)	92.6 (88.6–96.7)	**1.78 (1.14–2.78)**		0.40 (0.16–1.00)	
**Ethnicity**							
	Dutch overseas territories ^d^ and Suriname	857 (79.7)	91.9 (90.1–93.8)	Ref.	**0.0001**		
	Indigenous Dutch and other Western countries	80 (7.4)	92.5 (86.7–98.3)	1.19 (0.47–3.00)			
	Latin America and other non-Western countries	138 (12.8)	81.9 (75.4–88.3)	**3.38 (1.93–5.90)**			
**(Maternal) educational level ^e^**					**0.097**		
	High	171 (15.9)	90.6 (86.3–95.0)	Ref.			
	Middle	358 (33.3)	92.5 (89.7–95.2)	1.02 (0.50–2.10)			
	Low	479 (44.6)	89.4 (86.6–92.1)	1.89 (0.95–3.75)			
	Unknown	67 (6.2)	91.0 (84.2–97.9)	1.49 (0.52–4.29)			
**Monthly gross income household**					0.74		
	High (≥$3,001)	187 (17.4)	91.4 (87.4–95.5)	Ref.			
	Middle ($1501–3000)	272 (25.3)	91.2 (87.8–94.6)	1.16 (0.57–2.36)			
	Low (≤$1500)	219 (20.4)	89.0 (84.9–93.2)	1.37 (0.67–2.81)			
	Does not want to answer	144 (13.4)	88.9 (83.7–94.0)	1.70 (0.77–3.75)			
	Unknown	253 (23.5)	92.1 (88.8–95.4)	1.26 (0.56–2.85)			
**Resident of Caribbean Netherlands since, years of age**					**<0.0001**		**0.005**
	0–1	703 (65.4)	91.8 (89.7–93.8)	Ref.		Ref.	
	2–10	144 (13.4)	93.1 (88.9–97.2)	1.45 (0.69–3.02)		1.11 (0.52–12.25)	
	11–17	47 (4.4)	72.3 (59.5–85.1)	**9.17 (4.07–20.7)**		**5.12 (2.13–12.30)**	
	18–41	128 (11.9)	89.8 (84.6–95.1)	1.89 (0.84–4.27)		1.03 (0.45–2.37)	
	Unknown	53 (4.9)	88.7 (80.1–97.2)	1.85 (0.70–4.90)		1.46 (0.54–3.97)	
**Number of vaccinations against measles ^f^**					**<0.0001**		**<0.0001**
	2 or more doses	349 (32.5)	95.1 (92.9–97.4)	Ref.		Ref.	
	1 dose	416 (38.7)	94.7 (92.6–96.9)	0.79 (0.38–1.61)		0.74 (0.36–1.52)	
	(Partly) followed NIP (as a child) (self-reported)	213 (19.8)	85.9 (81.2–90.6)	**4.29 (2.13–8.63)**		**3.24 (1.55–6.75)**	
	Not vaccinated	97 (9.0)	68.0 (58.7–77.3)	**6.82 (3.21–14.49)**		**5.67 (2.62–12.25)**	
**(Parent/caregiver) influenced by beliefs about vaccination ^g^**					0.83		
	Yes	118 (11.0)	89.0 (83.3–94.6)	Ref.			
	No	820 (76.3)	90.7 (88.7–92.7)	0.83 (0.43–1.60)			
	Unknown	137 (12.7)	92.0 (87.4–96.5)	0.79 (0.32–1.92)			
**Household size, persons**					0.97		
	Single-person household	48 (4.5)	89.6 (80.9–98.2)	Ref.			
	2–5	868 (80.7)	90.9 (89.0–92.8)	0.86 (0.31–2.35)			
	≥6	150 (14.0)	90.0 (85.2–94.8)	0.98 (0.31–3.08)			
	Unknown	9 (0.8)	88.9 (68.3–100.0)				
**Contact yesterday, persons**					0.31		
	0–8	415 (38.6)	89.6 (86.7–92.6)	Ref.			
	≥9	537 (50.0)	92.2 (89.9–94.5)	0.89 (0.55–1.44)			
	Unknown	123 (11.4)	87.8 (82.0–93.6)	1.49 (0.77–2.90)			

^a^ On Bonaire, NIP eligible participants for measles include those until 41 y (in accordance with data from Curaçao), and on St. Eustatius and Saba until 35 y. ^b^ Crude odds ratios were adjusted for sex and age, and significant (a)odds ratios (ORs) were marked in bold type. ^c^
*p* values were determined by means of Wald tests for logistic regression, and significant *p* values (<0.1 in univariate and <0.05 in multivariate analysis) were marked in bold type. ^d^ Dutch overseas territories include the islands: Bonaire, Saba and St. Eustatius (i.e., Caribbean Netherlands), and Aruba, Curaçao and St. Maarten. ^e^ Maternal educational level was used for participants 0–11 y, active education was used for participants 12–25 y and highest accomplished educational level was used for participants >25 y. Low = no education, primary school, pre-vocational education (VMBO), lower vocational education (LBO/MBO-1) and lower general secondary education (MAVO/VMBO). Middle = intermediate/secondary vocational education (MBO-2-4), higher/senior vocational education (HAVO) and pre-university education (VWO/Gymnasium) and high = higher professional education (HBO), university BSc., university MSc. and doctorate. ^f^ The self-reported variable on NIP participation was used if a vaccination certificate was unavailable. Participants were categorized as ‘not vaccinated’ if both a vaccination certificate was unavailable as well as if they self-reported about no participation in the NIP or did not know whether they participated. ^g^ Beliefs include anthroposophy and natural healing, religion, social media, and other. Abbreviations: aOR, adjusted odds ratio; CI, confidence interval; OR, odds ratio; Ref., reference category.

**Table 4 vaccines-07-00137-t004:** Risk factor analysis for mumps IgG seronegativity among the total Health Study Caribbean Netherlands population with a blood sample and questionnaire data.

Potential Risk Factor for Mumps Seronegativity		*n* (%) *n* = 1816	% Mumps Seropositive (95% CI)	Univariate Crude OR ^a^ (95% CI)	*p* Value ^b^	Multivariate aOR ^a^ (95% CI)	*p* Value ^b^
**Island**					0.29		
	Bonaire	1128 (62.1)	85.2 (83.1–87.3)	Ref.			
	St. Eustatius	467 (25.7)	84.2 (80.8–87.5)	1.04 (0.77–1.42)			
	Saba	221 (12.2)	80.5 (75.3–85.8)	1.36 (0.93–1.99)			
**Sex**					0.25		0.19
	Men	818 (45.0)	83.5 (81.0–86.0)	1.17 (0.90–1.52)		1.20 (0.92–1.56)	
	Women	998 (55.0)	85.1 (82.9–87.3)	Ref.		Ref.	
**Age group, years**					**<0.0001**		**<0.0001**
	0–1	49 (2.7)	36.7 (23.3–50.2)	**17.07 (8.57–34.00)**		**10.15 (4.72–21.80)**	
	2–10	356 (19.6)	83.1 (79.3–87.0)	**1.98 (1.25–3.15)**		**1.87 (1.11–3.17)**	
	11–17	335 (18.5)	90.7 (87.6–93.9)	Ref.		Ref.	
	18–29	172 (9.5)	84.3 (78.9–89.7)	**1.86 (1.07–3.24)**		1.81 (0.99–3.32)	
	30–59	493 (27.1)	83.2 (79.9–86.5)	**2.02 (1.30–3.14)**		2.02 (0.97–4.21)	
	60–90	411 (22.6)	87.3 (84.1–90.6)	1.43 (0.89–2.29)		1.44 (0.62–4.86)	
**Ethnicity**					0.13		
	Dutch overseas territories ^c^ and Suriname	1312 (72.2)	83.5 (81.5–85.5)	Ref.			
	Indigenous Dutch and other Western countries	223 (12.3)	83.4 (78.5–88.3)	1.03 (0.69–1.53)			
	Latin America and other non-Western countries	281 (15.5)	89.0 (85.3–92.6)	0.66 (0.44–1.00)			
**(Maternal) educational level ^d^**					0.39		
	High	326 (18.0)	81.9 (77.7–86.1)	Ref.			
	Middle	466 (25.7)	85.6 (82.4–88.8)	0.83 (0.55–1.24)			
	Low	877 (48.3)	84.3 (81.9–86.7)	1.10 (0.77–1.57)			
	Unknown	147 (8.1)	86.4 (80.8–91.9)	0.88 (0.50–1.54)			
**Monthly gross income**					0.61		
	High (≥$3001)	346 (19.0)	81.5 (77.4–85.6)	Ref.			
	Middle ($1501–3000)	475 (26.2)	84.6 (81.4-87.9)	0.81 (0.55-1.18)			
	Low (≤$1500)	513 (28.2)	85.2 (82.1–88.3)	0.84 (0.57–1.22)			
	Does not want to answer	199 (11.0)	83.4 (78.2–88.6)	1.08 (0.67–1.75)			
	Unknown	283 (15.6)	86.6 (82.6–90.5)	1.04 (0.62–1.73)			
**Resident of Caribbean Netherlands since, years of age**					**0.009**		**0.02**
	0–1	1034 (56.9)	83.7 (81.4–85.9)	Ref.		Ref.	
	2–10	161 (8.9)	82.0 (76.0–87.9)	1.54 (0.98–2.43)		1.47 (0.92–2.34)	
	11–17	54 (3.0)	78.9 (64.5–87.3)	**2.85 (1.42–5.71)**		**2.12 (1.02–4.41)**	
	18–39	294 (16.2)	86.7 (82.9–90.6)	0.73 (0.48–1.11)		0.63 (0.41–0.97)	
	40–59	163 (9.0)	89.0 (84.1–93.8)	0.63 (0.36–1.09)		0.61 (0.35–1.06)	
	≥60	28 (1.5)	89.3 (77.8–100.0)	0.70 (0.20–2.44)		0.68 (0.20–2.35)	
	Unknown	82 (4.5)	84.1 (76.2–92.1)	1.00 (0.53–1.89)		0.95 (0.50–1.81)	
**Number of vaccinations against mumps ^e^**					**<0.0001**		**<0.0001**
	2 or more doses	349 (19.1)	94.3 (91.8–96.7)	Ref.		Ref.	
	1 dose	421 (23.0)	81.0 (77.2–84.7)	**2.81 (1.63–4.86)**		**2.82 (1.63–4.86)**	
	(Partly) followed NIP (as a child) (self-reported)	163 (8.9)	78.5 (72.2–84.8)	**3.93 (2.08–7.42)**		**3.75 (1.96–7.15)**	
	Not vaccinated	79 (4.3)	58.2 (47.3–69.1)	**7.17 (3.56–14.44)**		**7.00 (3.45–14.18)**	
	Not eligible for NIP	817 (44.7)	85.7 (83.3–88.1)	**3.06 (1.39–6.76)**		**2.93 (1.32–6.49)**	
**Influenced by beliefs about vaccination ^f^**					0.35		
	Yes	195 (10.7)	84.1 (79.0–89.2)	Ref.			
	No	1369 (75.4)	84.7 (82.8–86.6)	1.00 (0.66–1.54)			
	Unknown	252 (13.9)	82.5 (77.8–87.2)	1.31 (0.78–2.21)			
**Household size, persons**					0.96		
	Single-person household	218 (12.0)	85.5 (80.9–90.2)	Ref.			
	2–5	1382 (76.1)	84.2 (82.3–86.2)	1.01 (0.66–1.54)			
	≥6	204 (11.2)	84.8 (79.9–89.7)	0.95 (0.54–1.68)			
	Unknown	12 (0.7)	75.0 (50.5–99.5)	1.40 (0.33–5.92)			
**Contact yesterday, persons**					0.52		
	0–8	810 (44.6)	84.4 (81.9–86.9)	Ref.			
	≥9	794 (43.7)	84.8 (82.3–87.3)	0.99 (0.75–1.32)			
	Unknown	212 (11.7)	82.5 (77.4–87.7)	1.25 (0.82–1.89)			
**Mumps symptoms in preceding year ^g^**					0.43		
	Yes	27 (1.5)	81.5 (66.8–96.1)	Ref.			
	No	1622 (89.3)	84.6 (82.9–86.4)	0.68 (0.25–1.82)			
	Unknown	167 (9.2)	82.0 (76.2–87.9)	0.85 (0.30–2.47)			

^a^ Crude odds ratios were adjusted for sex and age, and significant (a)ORs are marked in bold type. ^b^
*p* values were determined by means of Wald tests for logistic regression, and significant *p* values (<0.1 in univariate and <0.05 in multivariate analysis) were marked in bold type. ^c^ Dutch overseas territories include the islands: Bonaire, Saba and St. Eustatius (i.e., Caribbean Netherlands), and Aruba, Curaçao and St. Maarten. ^d^ Maternal educational level was used for participants 0–11 y, active education was used for participants 12–25 y and highest accomplished educational level was used for participants >25 y. Low = no education, primary school, pre-vocational education (VMBO), lower vocational education (LBO/MBO-1) and lower general secondary education (MAVO/VMBO). Middle = intermediate/secondary vocational education (MBO-2-4), higher/senior vocational education (HAVO) and pre-university education (VWO/Gymnasium); and high = higher professional education (HBO), university BSc., university MSc. and doctorate. ^e^ The self-reported variable on NIP participation was used if a vaccination certificate was unavailable. Participants were categorized as ‘not vaccinated’ if both a vaccination certificate was unavailable as well as if they self-reported about no participation in the NIP or did not know whether they participated. On Bonaire, NIP eligible participants for mumps include those until 36 y (in accordance with data from Curaçao), on St. Eustatius until 29 y and on Saba until 34 y. ^f^ Beliefs include anthroposophy and natural healing, religion, social media and other. ^g^ Whether or not diagnosed by a physician. Abbreviations: aOR, adjusted odds ratio; CI, confidence interval; OR, odds ratio; Ref., reference category.

**Table 5 vaccines-07-00137-t005:** Risk factor analysis for rubella IgG seronegativity among the total Health Study Caribbean Netherlands population with a blood sample and questionnaire data.

Potential Risk Factor for Rubella Seronegativity		*n* (%) *n* = 1816	% Rubella Seropositive (95% CI)	Univariate Crude OR ^a^(95% CI)	*p* Value ^b^	Multivariate aOR ^a^ (95% CI)	*p* Value ^b^
**Island**					0.48		
	Bonaire	1128 (62.1)	86.1 (84.1–88.1)	Ref.			
	St. Eustatius	467 (25.7)	85.9 (82.7–89.0)	1.07 (0.77–1.48)			
	Saba	221 (12.2)	81.9 (76.8–87.0)	1.28 (0.86–1.90)			
**Sex**					0.23		0.34
	Men	818 (45.0)	87.0 (84.7–89.3)	Ref.		0.87 (0.65–1.16)	
	Women	998 (55.0)	84.3 (82.0–86.5)	1.18 (0.90–1.56)		Ref.	
**Age group, years**					**<0.0001**		**<0.0001**
	0–1	49 (2.7)	44.9 (31.0–58.8)	**24.28 (11.53–51.14)**		**14.60 (6.50–32.81)**	
	2–10	356 (19.6)	95.2 (93.0–97.4)	Ref.		Ref.	
	11–17	335 (18.4)	93.7 (91.1–96.3)	1.33 (0.69–2.57)		1.12 (0.54–2.31)	
	18–39	337 (18.6)	85.5 (81.7–89.2)	**3.30 (1.86–5.87)**		**2.61 (1.33–5.10)**	
	40–59	328 (18.1)	79.6 (75.2–83.9)	**5.04 (2.89–8.79)**		**3.34 (1.52–7.35)**	
	60–90	411 (22.6)	80.0 (76.2–83.9)	**4.93 (2.86–8.50)**		**2.89 (1.25–6.64)**	
**Ethnicity**					**0.0002**		
	Dutch overseas territories ^c^ and Suriname	1312 (72.2)	84.8 (82.8–86.7)	**3.16 (1.83–5.43)**			
	Indigenous Dutch and other Western countries	223 (12.3)	92.8 (89.4–96.2)	Ref.			
	Latin America and other non-Western countries	281 (15.5)	83.3 (78.9–87.6)	**2.94 (1.60–5.39)**			
**(Maternal) educational level ^d^**					**0.053**		
	High	326 (18.0)	86.2 (82.4–89.9)	Ref.			
	Middle	466 (25.7)	87.8 (84.8–90.7)	1.18 (0.76–1.84)			
	Low	877 (48.3)	83.8 (81.4–86.2)	**1.59 (1.08–2.35)**			
	Unknown	147 (8.1)	87.1 (81.6–92.5)	1.04 (0.57–1.89)			
**Monthly gross income**					0.22		
	High (≥$3001)	346 (19.0)	87.6 (84.1–91.1)	Ref.			
	Middle ($1501–3000)	475 (26.2)	85.5 (82.3–88.6)	1.28 (0.84–1.96)			
	Low (≤$1,501)	513 (28.2)	80.9 (77.5–84.3)	**1.61 (1.07–2.41)**			
	Does not want to answer	199 (11.0)	85.9 (81.1–90.8)	1.55 (0.90–2.65)			
	Missing	283 (15.6)	91.2 (87.9–94.5)	1.27 (0.69–2.33)			
**Resident of Caribbean Netherlands since, years of age**					**<0.0001**		**<0.0001**
	0–1	1034 (56.9)	84.0 (81.8–86.3)	**3.44 (1.96–6.05)**		**3.58 (2.03–6.29)**	
	2–10	161 (8.9)	87.0 (81.7–92.2)	**5.68 (2.68–12.03)**		**5.48 (2.57–11.69)**	
	11–17	54 (3.0)	79.6 (68.9–90.4)	**8.70 (3.46–21.91)**		**6.83 (2.66–17.52)**	
	18–39	294 (16.2)	86.1 (82.1–90.0)	1.88 (1.00–3.51)		1.72 (0.91–3.23)	
	40–59	163 (9.0)	90.2 (85.6–94.8)	Ref.		Ref.	
	≥60	28 (1.5)	96.4 (89.5–100.0)	0.36 (0.05–2.83)		0.37 (0.05–2.91)	
	Unknown	82 (4.5)	90.2 (83.8–96.7)	1.76 (0.70–4.43)		1.63 (0.64–4.11)	
**Number of vaccinations against rubella ^e^**					**<0.0001**		**<0.0001**
	2 or more doses	349 (19.2)	95.1 (92.9–97.4)	Ref.		Ref.	
	1 dose	425 (23.4)	92.7 (90.2–95.2)	1.27 (0.65–2.48)		1.30 (0.67–2.53)	
	(Partly) followed NIP (as a child) (self-reported)	245 (13.5)	84.9 (80.4–89.4)	**2.37 (1.21–4.65)**		**2.45 (1.24–4.84)**	
	Not vaccinated	101 (5.6)	66.3 (57.1–75.6)	**4.76 (2.32–9.75)**		**5.15 (2.50–10.62)**	
	Not eligible for NIP	696 (38.3)	79.3 (76.3–82.3)	**3.68 (1.74–7.79)**		**3.75 (1.77–7.96)**	
**Influenced by beliefs about vaccination ^f^**					0.99		
	Yes	195 (10.7)	85.1 (80.1–90.1)	Ref.			
	No	1369 (75.4)	85.5 (83.7–87.4)	1.02 (0.65–1.58)			
	Unknown	252 (13.9)	85.7 (81.4–90.0)	1.04 (0.60–1.81)			
**Household size, persons**					0.52		
	Single-person household	218 (12.0)	78.4 (73.0–83.9)	Ref.			
	2–5	1382 (76.1)	86.5 (84.7–88.3)	0.78 (0.53–1.13)			
	≥6	204 (11.2)	86.3 (81.5–91.0)	0.88 (0.51–1.52)			
	Unknown	12 (0.7)	91.7(76.0–100.0)	0.40 (0.04–3.74)			
**Contact yesterday, persons**					**0.003**		
	0–8	810 (44.6)	84.3 (81.8–86.8)	Ref.			
	≥9	794 (43.7)	88.4 (86.2–90.6)	0.85 (0.63–1.15)			
	Unknown	212 (11.7)	79.2 (73.8–84.7)	**1.76 (1.17–2.63)**			

^a^ Crude odds ratios were adjusted for sex and age, and significant (a)ORs are marked in bold type. ^b^
*p* values were determined by means of Wald tests for logistic regression, and significant *p* values (<0.1 in univariate and <0.05 in multivariate analysis) were marked in bold type. ^c^ Dutch overseas territories include the islands: Bonaire, Saba and St. Eustatius (i.e., Caribbean Netherlands), and Aruba, Curaçao and St. Maarten. ^d^ Maternal educational level was used for participants 0–11 y, active education was used for participants 12–25 y and highest accomplished educational level was used for participants >25 y. Low = no education, primary school, pre-vocational education (VMBO), lower vocational education (LBO/MBO-1) and lower general secondary education (MAVO/VMBO). Middle = intermediate/secondary vocational education (MBO-2-4), higher/senior vocational education (HAVO) and pre-university education (VWO/Gymnasium) and high = higher professional education (HBO), university BSc., university MSc. and doctorate. ^e^ The self-reported variable on NIP participation was used if a vaccination certificate was unavailable. Participants were categorized as ‘not vaccinated’ if both a vaccination certificate was unavailable as well as if they self-reported about no participation in the NIP or did not know whether they participated. On Bonaire, NIP eligible participants for mumps include those until 36 y (in accordance with data from Curaçao), on St. Eustatius until 29 y and on Saba until 34 y. ^f^ Beliefs include anthroposophy and natural healing, religion, social media and other. Abbreviations: aOR, adjusted odds ratio; CI, confidence interval; OR, odds ratio; Ref., reference category.

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
