# Peer review of "Seroepidemiology of Measles, Mumps and Rubella on Bonaire, St. Eustatius and Saba: The First Population-Based Serosurveillance Study in Caribbean Netherlands"

_vaccines, 2019, doi:10.3390/vaccines7040137_

Round 1

Reviewer 1 Report

This work by authors is   General comments- Most of the manuscript is well written, however there were few grammatical mistakes.

Specific comments-

Line 78- study design and study population. - As this is a large population based study, a flow chart or work flow need to be summarized. This will help the readers in better understanding of the results and discussion. Line 79- what was the criteria in age grouping? The number of years in each groups is not uniform. Could this lead to skewing of results and it’s interpretation? Line 81- total cases invited were 7768 and following sentence mentions, consent was obtained from all participants. However in results, line 144, 1990 participants were included. After repeated reading and continuing to further sections of the manuscript, then only its becoming clear. As said above, a flow chart would help in better understanding. Line 82- Shouldn’t be the age >/= 18 years? Line 89- were medical records available for all the patients without vaccination certificate? If not how was the data correlated? Line 97- how was the dilution calculated to 1:200? Could authors briefly describe the calculation or cite a reference?

Reviewer 2 Report

Lines 403-405: In your results, include the data that shows that storage and transportation had no effect, since you state that you investigated this and found no effect.

Lines 403-405: “…we could not exclude the possible effect of storage and transportation on the antibody levels. However, our data showed minimal overall effect, indicating that storage and transportation had no significantly effect on our results.”

Line 409: replace indicates with “indicate”

Line 415: replace “base” with “basis”
